# Establishment of an In Vitro Model of Pseudorabies Virus Latency and Reactivation and Identification of Key Viral Latency-Associated Genes

**DOI:** 10.3390/v15030808

**Published:** 2023-03-22

**Authors:** Li Pan, Mingzhi Li, Xinyu Zhang, Yu Xia, Assad Moon Mian, Hongxia Wu, Yuan Sun, Hua-Ji Qiu

**Affiliations:** 1State Key Laboratory of Veterinary Biotechnology, Harbin Veterinary Research Institute, Chinese Academy of Agricultural Sciences, 678 Haping Road, Harbin 150069, Chinasunyuan@caas.cn (Y.S.); 2School of Animal Science and Technology, Henan Institute of Science and Technology, Xinxiang 453003, China; 3School of Life Science and Engineering, Foshan University, Foshan 528225, China

**Keywords:** pseudorabies virus, latent infection, reactivation, thermoregulation, in vitro model, *UL54*

## Abstract

Alphaherpesviruses infect humans and most animals. They can cause severe morbidity and mortality. The pseudorabies virus (PRV) is a neurotropic alphaherpesvirus that can infect most mammals. The PRV persists in the host by establishing a latent infection, and stressful stimuli can induce the latent viruses to reactivate and cause recurrent diseases. The current strategies of antiviral drug therapy and vaccine immunization are ineffective in eliminating these viruses from the infected host. Moreover, overspecialized and complex models are also a major obstacle to the elucidation of the mechanisms involved in the latency and reactivation of the PRV. Here, we present a streamlined model of the latent infection and reactivation of the PRV. A latent infection established in N2a cells infected with the PRV at a low multiplicity of infection (MOI) and maintained at 42 °C. The latent PRV was reactivated when the infected cells were transferred to 37 °C for 12 to 72 h. When the above process was repeated with a *UL54*-deleted PRV mutant, it was observed that the *UL54* deletion did not affect viral latency. However, viral reactivation was limited and delayed. This study establishes a powerful and streamlined model to simulate PRV latency and reveals the potential role of temperature in PRV reactivation and disease. Meanwhile, the key role of the early gene *UL54* in the latency and reactivation of PRV was initially elucidated.

## 1. Introduction

The pseudorabies virus (PRV) belongs to the subfamily *Alphaherpesvirinae* of the family *Herpesviridae* and causes a lethal viral disease in pigs, which causes severe economic losses to the swine industry around the world [1,2]. PRV is a neuroinvasive herpesvirus that causes various infections from acute to lifelong latent infections of the peripheral nervous system (PNS) [3]. However, the molecular mechanisms underlying the establishment of alphaherpesvirus latency and reactivation remain poorly understood [4,5,6]. For herpesviruses, primary infection begins with the productive infection of mucosal epithelial cells. In less than a day, each infected cell will produce thousands of progeny [7]. It is generally accepted that the latency of alphaherpesviruses is characterized by the episomal maintenance of the viral genome and by a state of chronic infection that does not lead to the production of infectious viral particles [8]. When known stress signaling and neuronal survival pathways are activated, the viral genome is transcribed and replicated, producing new viral particles that travel back to the original infection site to ensure the spread of infection to other hosts [9]. Chronic viruses, such as herpesviruses, influence host physiology and have evolved to use inflammation to regulate viral latency and reactivation and to modulate the inflammatory state of the immune system [10]. The incubation of cultures at a lower temperature (34 °C) to induce the increased reactivation of the varicella zoster virus (VZV) has been reported, resulting in diffuse productive infection [11]. There is a hypothesis that higher temperatures may be conducive to the silent state of the VZV. An elevated core temperature may limit the spread of the virus at the onset of latency and the reactivation events. This temperature effect may be due to viral enzyme activity, lower-temperature-initiated pathways, or reduced innate antiviral response efficiency [11,12].

Gene expression in alphaherpesvirus lytic infections follows a sequential order: immediate early (IE), early (E), and late (L) genes [13]. The UL54 protein is an early PRV protein with RNA binding activity that regulates the expression of late genes, which is homologous to the ICP27 of HSV-1 [14,15]. The ICP27 protein is a key molecule in sensing environmental changes [16]. Based on this, the present study hypothesized that UL54 may have an essential role in the latency and reactivation of PRV. Even though PRV is one of the best-studied viral pathogens, the regulation of latency and reactivation is not well understood. This is due to several challenges, including difficulty modeling latency in vivo and in vitro and a limited understanding of cell biology. Furthermore, modeling techniques are unavailable to the broader scientific community due to high requirements, high costs, and a lack of ethical approval in some countries.

As temperature plays a crucial role in the latent infection and reactivation of alphaherpesviruses, we report for the first time the use of temperature for the precise control of the latent infection and reactivation of the PRV in mouse neuroma cells (N2a). We believe the model simulates the in vivo microenvironment of cells in the latent phase of viral infection under in vitro conditions. This model has been used to explore the fundamental aspects of the PRV lifecycle and will promote the development of this area.

## 2. Materials and Methods

### 2.1. Cells and Viruses

A mouse-derived neuroblastoma cell line (N2a), baby hamster kidney 21 (BHK-21) cells, and porcine kidney 15 (PK-15) cells were cultured in DMEM supplemented with 10% fetal bovine serum (FBS). The wild-type PRV strain (WT-PRV) (GenBank accession number: KJ789182.1) was isolated and identified in our laboratory [17]. The fosmid library of WT-PRV was used for the construction of the recombinant PRV mutants [18]. The reporter gene encoding EGFP was inserted as an additional reading frame between the US9 and US2 reading frames to construct the recombinant rPRV-EGFP. We then constructed a recombinant rPRV-ΔUL54-EGFP with the *UL54* gene deleted based on rPRV-EGFP. These two PRV mutants were rescued in BHK-21 cells as described previously [18]. The primers used to construct the recombinant viruses are listed in Table 1.

### 2.2. One-Step and Multistep Growth Curves and Plaque Assay

For one-step growth curves, PK-15 cells in 24-well cell culture plates were infected with WT-PRV or rPRV-EGFP (MOI = 5) and were incubated for 2 h, and they were then washed with PBS to clear the uninfected viral particles and were cultured in fresh medium. The cell supernatants were harvested at different time points. For multistep growth curves, PK-15 cells were infected with WT-PRV or rPRV-ΔUL54-EGFP (MOI = 0.01). Growth curves were plotted based on median tissue culture infective doses (TCID_50_) or the viral DNA copies of all the samples [19]. Plaque assay was performed as described previously [20].

### 2.3. PCR, qPCR, and RT-qPCR

The genomic DNA of WT-PRV, rPRV-EGFP, and rPRV-ΔUL54-EGFP was extracted and used as the template for PCR kit (catalog no. AG11510; Agbio, Changsha, China) or qPCR kit (catalog no. Q311; Vazyme, Beijing, China). Total RNA was isolated from the infected or uninfected cells, and RT-qPCR kit (catalog no. A336; Genstar, Beijing, China) was used to quantify viral mRNA. The tests were performed according to the kit instructions. The primers for PCR, qPCR, and RT-qPCR are listed in Table 2.

### 2.4. Transmission Electronic Microscopy 

The samples were prepared as reported previously [18]. Cell culture medium was centrifuged at 3000× *g* for 10 min, the supernatant was collected and centrifuged at 10,000× *g* for 10 min, and then the pellet was resuspended in PBS. The samples were negatively stained with 2% phosphotungstic acid, and the morphology of viral particles was observed under transmission electronic microscopy (TEM).

### 2.5. Western Blotting 

The N2a cells infected with the recombinant viruses of different MOIs were cultured at 37 and 42 °C for 120 h. At 120 h postinfection (hpi), the cells were harvested, and samples were prepared for Western blotting. The mouse anti-gB and -gD monoclonal antibodies (MAbs) were from our laboratory. Anti-β-actin MAb was commercially purchased from Sigma (catalog no. A1978; Sigma, Shanghai, China). Goat antimouse FITC antibodies (catalog no. sc-516140; Sigma, Shanghai, China) were used as secondary antibodies. Total protein extraction and Western blotting analysis were carried out following the previously described protocols [20]. 

### 2.6. Thermoregulation Model of Latent Infection and Reactivation of PRV

To establish a model of latent PRV infection, N2a, BHK-21, and PK-15 cells were seeded separately into 35 mm dishes at 5 × 10^5^ per dish and were cultured with a complete medium. For infection, in brief, rPRV-EGFP was added to the three cell types mentioned above at MOIs of 1.0, 0.1, 0.01, and 0.001, respectively. The infected cells were incubated continuously at 37 or 42 °C for 72 or 120 h. Fluorescence was monitored at 72 hpi to investigate the optimal cell type and MOI for establishing PRV latent infection. N2a cells were infected with rPRV-EGFP (MOI = 0.01) and were incubated at 37 °C for different amounts of time to determine the optimal incubation time. To determine the incubation time for noninfectious PRV replication, the cells were then transferred to 42 °C for 120 h, and EGFP fluorescence was monitored. Subsequently, latent infection of PRV was established with optimized parameters and was maintained at 42 °C for 120 h (i.e., N2a cells were infected with rPRV-EGFP at an MOI of 0.01, were incubated at 37 °C for 2 h, and were then transferred to 42 °C). In addition, qPCR, RT-qPCR, and Western blotting were used to analyze viral DNA, mRNA, and proteins in the samples. For the reactivation of latent PRV, N2a cells were transferred from 42 to 37 °C for mild hypothermic incubation (cold stress), activating the latent PRV. At 12, 24, 48, and 72 h postreactivation (hpr), the infectious PRV was measured through EGFP fluorescence and RT-qPCR. It should be noted that the medium was changed on alternate days throughout the model experiment to maintain normal cell growth.

For the experiments of latent infection and reactivation of rPRV-ΔUL54-EGFP, the experimental details were similar to the above procedures.

### 2.7. Statistical Analysis

Analysis of the data was performed using GraphPad Prism 8 statistical software. Student’s two-tailed *t*-test and two-way analysis of variance were used, and significant differences were defined when the *p*-value was <0.05.

## 3. Results

### 3.1. Generation and Identification of rPRV-EGFP

The recombinant PRV with the EGFP cassette was constructed using the PRV reverse genetic operating system established in our laboratory [18] (Figure 1A). The clone was identified as rPRV-EGFP after plaque purification and sequencing verification (Figure 1B). The size of the plaques produced by rPRV-EGFP was similar to that of WT-PRV (Figure 1C). Electron microscopy revealed that the rPRV-EGFP particles had an intact envelope similar to that of the parental virus (Figure 1D). Fluorescence imaging demonstrated the capability of rPRV-EGFP to express the EGFP protein (Figure 1E). The replication kinetics of rPRV-EGFP was very similar to that of the parental virus based on the one-step growth curve (Figure 1F). Overall, the recombinant PRV retained the parental phenotype, except for EGFP fluorescence.

### 3.2. Hyperthermic Stress Maintained PRV Latency in Neuron-like Cells

Inspired by the speculation of Markus et al. [11] and a real-world problem situation, we screened the optimal conditions for the thermoregulation model of the latent infection and reactivation of the PRV. In this study, we attempted to establish models of the latent infection and reactivation of PRV in PK-15 cells, BHK-21 cells, and N2a cells. The results showed that only the N2a cells allowed the PRV to establish a latent infection. The PRV with an MOI no more than 0.01 was able to maintain a stably latent infection at 42 °C (Figure 2).

Moreover, the study showed that one of the key factors determining the success or failure of the model was the incubation time at 37 °C. When the PRV was incubated at 37 °C for over 3 h, the lytic infection was not prevented, even when the cells were transferred to 42 °C (Figure 3A). Surprisingly, the virus (MOI = 0.01) was incubated at 37 °C for 2 h and was then moved to 37 or 42 °C for 120 h. No lytic viral replication was observed at 42 °C (Figure 3B). During the latency phase, the viral DNA was not replicated, but the latency-associated transcript (LAT) of PRV could be detected (Figure 3C); additionally, low levels of mRNA for IE180 (the immediate early gene of the PRV) were detected (Figure 3D). The cells infected with rPRV-EGFP at different MOIs and incubated at 37 or 42 °C for 120 h were harvested. These samples were used to detect the viral proteins gB and gD. The results showed that the virus did not produce lytic proteins at an MOI of 0.01 or at 42 °C (Figure 3E). The above results showed that, under hyperthermia stress, the viral DNA did not replicate and no viral proteins or viral particles was produced. However, the latent PRV only produced the LAT and low levels of IE180 transcripts.

### 3.3. Reduced Temperature Enhanced the Reactivation of the Latent PRV

It has been shown that mild hypothermia reactivates the VZV in latently infected neurons and enhances viral replication [11,21]. Therefore, we combined the treatment of persistent latent infection with hyperthermic stress (42 °C) and incubation with mild hypothermia (37 °C). According to the results, the virus was incubated at 42 °C for 120 h and was then transferred to 37 °C for additional incubation. Infectious virus particles were produced within 12 h (Figure 4A). The number of viral DNA copies increased with the viral reactivation time (Figure 4B). Upon reactivation, the expression levels of the late viral envelope proteins gB and gD were increased over time (Figure 4C). It was easy to see that hyperthermia stress drove the viruses into a quiescent state. At the same time, mild hypothermia could reactivate them to produce infectious viral particles and to infect other cells.

### 3.4. Deleting the UL54 Gene Affects Viral Particle Morphology and Replication Efficiency

As described in the Materials and Methods, the *UL54*-deleted mutant was constructed using rPRV-EGFP as the parental strain (Figure 5A). The same procedure as that described above was used to describe the phenotype of rPRV-ΔUL54-EGFP, which was produced through the fosmid system. Based on these results, the deletion of the *UL54* gene was confirmed through the PCR and sequencing of rPRV-ΔUL54-EGFP (Figure 5B). This further confirmed that the EGFP expression cassette was successfully inserted into the intended location in the PRV genome, and the green fluorescent protein was expressed (Figure 5C). Due to the *UL54* gene deletion, the plaques of rPRV-ΔUL54-EGFP were significantly smaller than those of WT-PRV (Figure 5D), and they lacked an outer membrane (Figure 5E) and replicated less efficiently compared to WT-PRV (Figure 5F). The results presented in this section demonstrate that the deletion of the *UL54* gene limited the expression of the viral envelope proteins, resulting in the production of fewer infectious viral particles.

### 3.5. Deleting the UL54 Gene Affects PRV Reactivation without Affecting Latency

To understand the effect of the *UL54* gene on the latency and reactivation of the PRV, rPRV-ΔUL54-EGFP was tested in the model developed above. The results showed that rPRV-ΔUL54-EGFP established a stably latent infection at 42 °C (Figure 6A). The characteristics of the latent rPRV-ΔUL54-EGFP were similar to those of rPRV-EGFP. The results showed that the rPRV-ΔUL54-EGFP infected group did not increase the DNA copies (Figure 6B), but it maintained higher levels of LAT and lower levels of IE180 transcripts (Figure 6C) and, last but not least, did not produce lytic viral proteins (Figure 6D). Subsequently, we transferred the cells from 42 to 37 °C. As expected, a few fluorescent spots appeared on the cells latently infected with rPRV-EGFP when incubated at 37 °C for 12 h. After 48 h incubation at 37 °C, the cells latently infected with rPRV-ΔUL54-EGFP showed insufficient fluorescent spots (Figure 6E). During reactivation, the DNA copies of rPRV-EGFP were significantly higher than rPRV-ΔUL54-EGFP during the same period, indicating that the deletion of the *UL54* gene significantly delayed viral reactivation (Figure 6F). As expected, the levels of viral lytic proteins were gradually increased with prolonged reactivation. The above data suggest that the early gene *UL54* plays an essential role in the reactivation of the PRV.

## 4. Discussion

In this study, we described an in vitro model of the latent infection and reactivation of the PRV through thermoregulation. Furthermore, based on this model, we described for the first time the essential role of the *UL54* gene in viral reactivation. To the best of our knowledge, this model is the most plausible and controllable model of the latent infection and reactivation of the PRV.

Alphaherpesviruses, the most common viruses in the world, can establish a lytic or latent infection in the peripheral nervous system of the host [22]. During the establishment of a latent infection, herpesviruses are maintained as epistomes in the nucleus of the host cell [23]. To elucidate the mechanisms of the latent infection and reactivation of alphaherpesviruses, several in vivo and in vitro models have been developed. The in vivo models include rhesus monkeys, domestic pigs, guinea pigs, and mice, and the in vitro models include primary neuronal cells, immortalized neuroma cells, and nonneuronal cells [11,12,13,22,24,25,26,27,28,29]. While most laboratories focus on their highly complex model systems and tools, the field would benefit from consolidating expertise and streamlining operational processes in a collaborative effort to address the global burden of alphaherpesviral diseases. Therefore, it is inevitable that the procedures of the model need to be simplified while ensuring its validity. Initial attempts were made to use the antiviral drug acyclovir to inhibit productive viral DNA replication in dissociated neurons to simulate culture latency. Since 1977, acyclovir has been a very effective antiherpetic drug [30]. The viral thymidine kinase in the infected cells activates the drug. Only after this activation does the drug interfere with viral DNA replication [31]. Pretreating neurons with acyclovir before viral infection prevents viral replication and forces productive infection into a dormant state, similar to latency in vivo [28]. This opened the way to a better understanding of the molecules stimulated by alphaherpesvirus reactivation. However, long-term administration cannot mimic the complex host response associated with in vivo infection. It is well known that, during infection, the ability of the host to mount an appropriate immune and inflammatory response against the pathogen is critical. Higher temperatures are expected to increase antiviral activity because fever enhances the inflammatory response to fight infection [32].

Our operational definition of experimental latency is the maintenance of viral genomes without virus production for prolonged periods that can be reactivated into productive viral infection, spreading to other cells. This definition distinguishes between latency and aborted, incomplete, or partial infection, which may apply to latency models in nonpermissive hosts or cell types that cannot support a full productive infection [11]. Understanding these factors that maintain latency and the driver of reactivation is important because understanding these processes may eventually allow us to target them to prevent reactivation. Latency involves a program of primarily suppressed regular lytic viral gene expression, leading to virion production. In addition to the previously mentioned details, we also made an initial attempt. According to the published studies, we also tried to use nonneuronal cells as models, including African green monkey kidney (Vero) cells, African green monkey embryonic kidney epithelial (MARC-145) cells, and Madin–Darby canine kidney (MDCK) cells. Unfortunately, a latent model could not be established in these cells. In addition, we tried different temperatures to establish latency. In 0.5 °C increments, the culture temperature of the cells was increased from 40 to 45 °C. Considering factors such as the physiological state of the cells and the reactivation of the virus, we selected 42 °C as the optimal incubation temperature. It has been reported that the PRV replicates rapidly in the mucosal or epithelial cells of pigs upon infection. Soon after, the intense inflammatory response induced through viral infection leads to a rapid rise in body temperature of up to 42 °C [33]. In the presence of a high temperature and the host immune defense, the residual virus enters the host’s peripheral nervous system to establish latency. Other studies have also shown that human herpes simplex virus 1 is able to maintain a latent state in human diploid fibroblasts and fetal lung cells cultured at 42 °C [34,35].

In conclusion, in the present study, in addition to the in vitro model of the PRV, we found that the *UL54* gene plays an essential role in the reactivation of the PRV. We believe that the *UL54* gene represents a potential target for latent virus clearance, and, given its importance, its exact function will be clarified in the future.

## Figures and Tables

**Figure 1 viruses-15-00808-f001:**
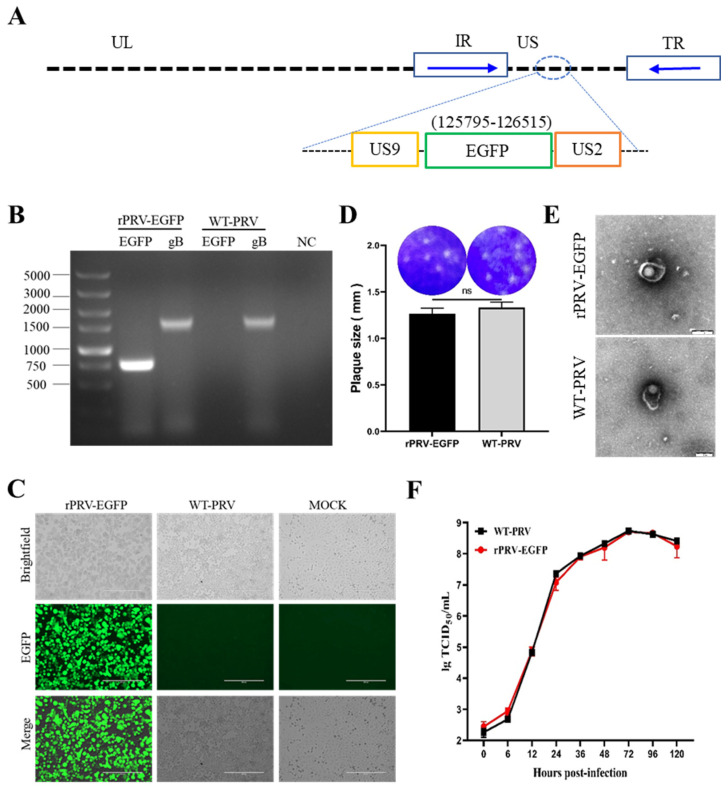
Generation and characterization of rPRV-EGFP. (**A**) Schematic diagram of the construction of the recombinant virus. Using the PRV genomic fosmid library, the EGFP sequence was inserted nondestructively downstream of the *US9* gene in the PRV genome. The relative position of the EGFP sequence in the WT-PRV genome is shown on top of the green box. (**B**) PCR amplification of the *EGFP* and *gB* genes from the genomes of rPRV-EGFP and WT-PRV. (**C**) The green fluorescence and cytopathic effects (CPEs) in the PK-15 cells infected with rPRV-EGFP and WT-PRV at 48 h postinfection (hpi). (**D**) Plaques of rPRV-EGFP and WT-PRV in the PK-15 cells. ns: not significant (*p* ≥ 0.05). The diameters of plaques were averaged for three independent experiments. ns: not significant. (**E**) Transmission electron microscopy photographs of rPRV-EGFP and WT-PRV. Scale bar = 200 nm. (**F**) One-step growth curves of rPRV-EGFP and WT-PRV in PK-15 cells.

**Figure 2 viruses-15-00808-f002:**
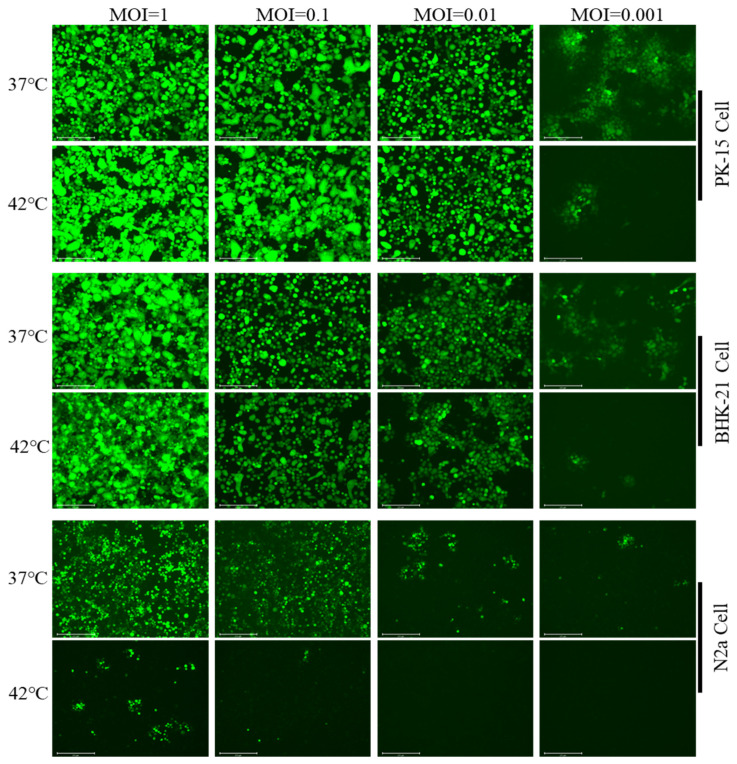
Screening for appropriate cell types and optimal infectious dose for in vitro PRV latent infection models. N2a, BHK-21, and PK-15 cells were seeded into 35 mm dishes at 5 × 10^5^ per dish and were cultured with complete medium. rPRV-EGFP was added to the three cell types at an MOI of 1, 0.1, 0.01, or 0.001. The infected cells were incubated continuously at 37 or 42 °C for 72 h. Fluorescence was monitored at 72 h postinfection.

**Figure 3 viruses-15-00808-f003:**
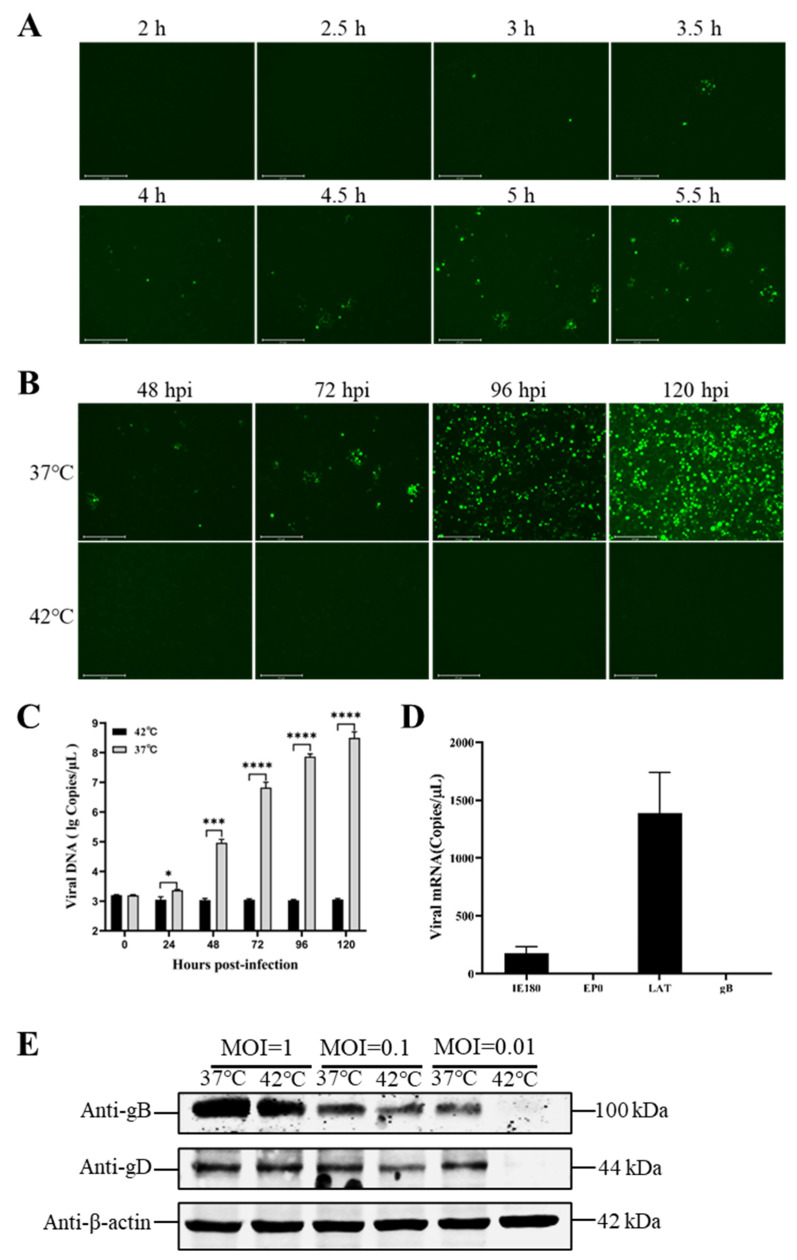
Optimal incubation time for screening and detection of latency indicators. (**A**) Optimal incubation time screening. N2a cells were infected with rPRV-EGFP (MOI = 0.01) and were incubated at 37 °C for 2, 2.5, 3, 3.5, 4, 4.5, 5, and 5.5 h. The cells were then transferred to 42 °C for 120 h, and EGFP fluorescence was monitored. (**B**) Identification of the optimal conditions for latent PRV infection. N2a cells were infected at an MOI of 0.01, were incubated at 37 °C for 2 h, then transferred to 42 °C for 120 h. (**C**) Latent and lytic viral DNA detection. N2a cells were infected with rPRV-EGFP at an MOI of 0.01 and were incubated at 37 °C for 2 h. After rPRV-EGFP was cultured at 42 °C for 120 h, the samples were then collected, and viral DNA was extracted; additionally, the *gB* gene was detected through real-time PCR to quantify genomic DNA. (**D**) Quantification of transcripts of latent viral genes. After rPRV-EGFP was cultured at 42 °C for 120 h, the samples were then collected, and the mRNAs of *IE180*, *EP0*, *LAT*, and *gB* were detected through RT-qPCR. (**E**) Detection of late viral protein expression levels at two culture temperatures. N2a cells were infected with different doses of rPRV-EGFP (MOI = 1, 0.1, or 0.01) and were incubated at 37 or 42 °C for 120 h, respectively. At 120 h postinfection (hpi), the cells were harvested, and samples were prepared for Western blotting. The homemade mouse anti-gB and -gD monoclonal antibodies were used as primary antibodies, and goat antimouse FITC antibodies were used as secondary antibodies. Bars represent the means ± SDs of three independent experiments. ns: not significant (*p* ≥ 0.05). * *p* < 0.05, *** *p* < 0.001, and **** *p* < 0.0001.

**Figure 4 viruses-15-00808-f004:**
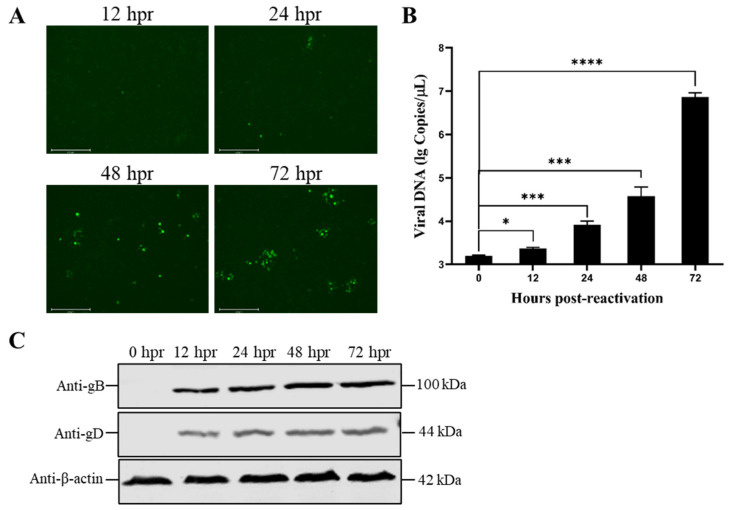
Reactivation of latent rPRV-EGFP. (**A**) Fluorescence spots of rPRV-EGFP during reactivation. N2a cells were infected with rPRV-EGFP (MOI = 0.01), were incubated at 42 °C for 120 h, and were then transferred to 37 °C for additional incubation. EGFP fluorescence was monitored at 12, 24, 48, and 72 h postreactivation (hpr). (**B**) Detection of viral genomic DNA during reactivation. The viral DNA was extracted, and the *gB* gene was detected at 0, 12, 24, 48, and 72 hpr. (**C**) Identification of expression levels of viral late proteins following rPRV-EGFP reactivation. N2a cells were infected with rPRV-EGFP (MOI = 0.01), were incubated at 42 °C for 120 h, and were then transferred to 37 °C. At 0, 12, 24, 48, and 72 hpr, the cells were harvested, and samples were prepared for Western blotting. The homemade mouse anti-gB and -gD monoclonal antibodies were used as primary antibodies, and goat antimouse FITC antibodies were used as secondary antibodies. Bars represent the means ± SDs of three independent experiments. ns: not significant (*p* ≥ 0.05). * *p* < 0.05, *** *p* < 0.001, and **** *p* < 0.0001.

**Figure 5 viruses-15-00808-f005:**
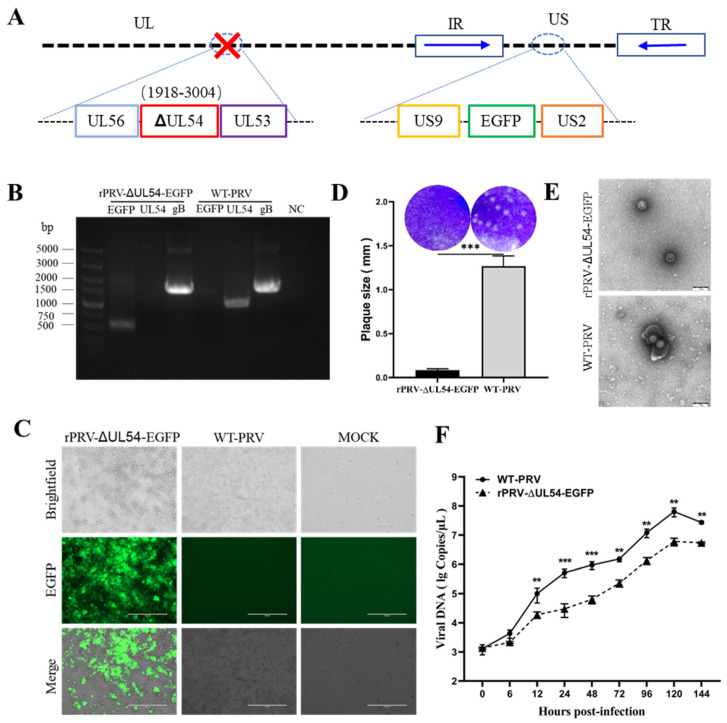
Generation and characterization of rPRV-ΔUL54-EGFP. (**A**) Schematic diagram of the construction of rPRV-ΔUL54-EGFP. Using the PRV genomic fosmid library, the *UL54* deletion mutant was constructed using rPRV-EGFP as the parental strain. The relative position of the *UL54* gene in the WT-PRV genome is shown on top of the red box. (**B**) PCR amplification of the *UL54*, *EGFP*, and *gB* genes from the genomes of rPRV-ΔUL54-EGFP and WT-PRV. (**C**) The green fluorescence and cytopathic effects (CPEs) in the PK-15 cells infected with rPRV-ΔUL54-EGFP and WT-PRV at 60 h postinfection (hpi). (**D**) Plaques of rPRV-ΔUL54-EGFP and WT-PRV in the PK-15 cells. The diameters of plaques were averaged for three independent experiments. (**E**) Transmission electron microscopy imagingof rPRV-ΔUL54-EGFP and WT-PRV. Scale bar = 200 nm. (**F**) Multistep growth curves of rPRV-ΔUL54-EGFP and WT-PRV in PK-15 cells. PK-15 cells were infected with rPRV-ΔUL54-EGFP and WT-PRV (MOI = 0.01). Bars represent the means ± SDs of three independent experiments. ns: not significant (*p* ≥ 0.05). ** *p* < 0.01, and *** *p* < 0.001.

**Figure 6 viruses-15-00808-f006:**
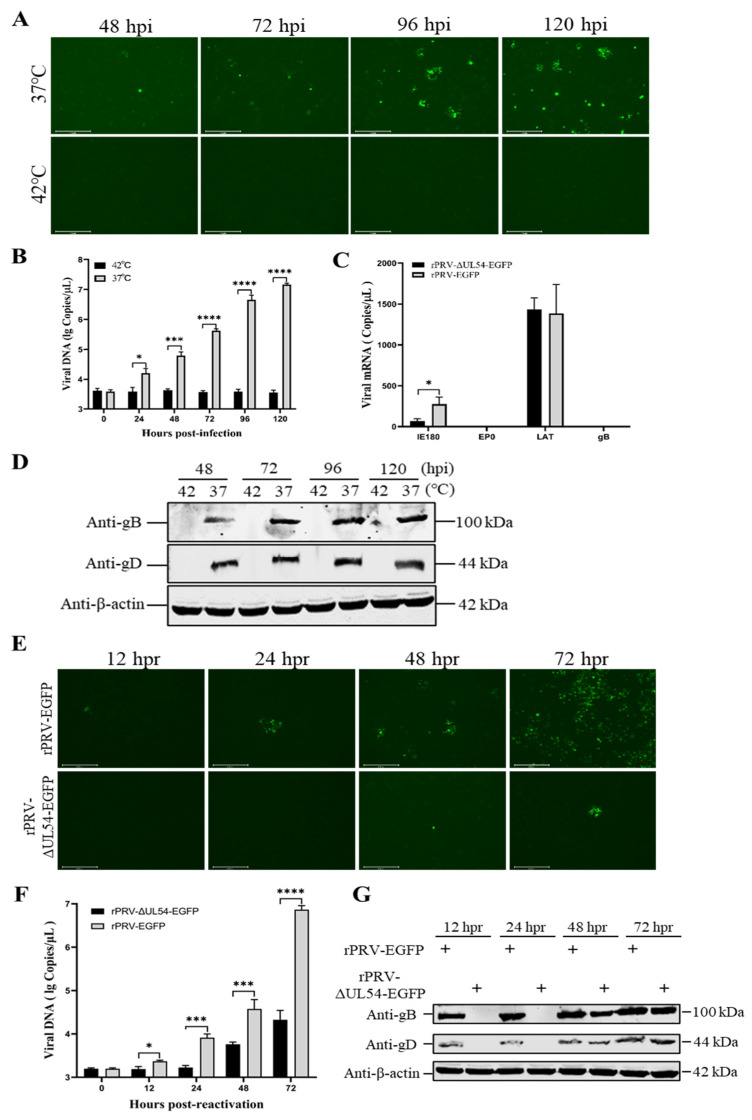
Latent infection and reactivation of rPRV-ΔUL54-EGFP. Based on the established model, rPRV-ΔUL54-EGFP was tested. (**A**) Identification of the effect of UL54 deletion on latent infection. (**B**) Latent and lytic viral DNA copy detection. N2a cells were infected with rPRV-ΔUL54-EGFP at an MOI of 0.01 and were incubated at 37 °C for 2 h. After rPRV-ΔUL54-EGFP was cultured at 42 °C for 120 h, the samples were then collected, and viral DNA was extracted; additionally, the *gB* gene was detected through real-time PCR to quantify genomic DNA. (**C**) Quantification of the transcripts of the latency-associated viral genes. After rPRV-ΔUL54-EGFP was cultured at 42 °C for 120 h, the samples were then collected, and total RNA was extracted; additionally, the mRNA of the *IE180*, *EP0*, *LAT*, and *gB* genes was detected through RT-qPCR. (**D**) Inability of latent rPRV-UL54-EGFP to express lytic viral proteins. (**E**) Fluorescence spots of rPRV-EGFP and rPRV-ΔUL54-EGFP during reactivation. N2a cells were infected with rPRV-EGFP and rPRV-ΔUL54-EGFP (MOI = 0.01), were incubated at 42 °C for 120 h, and were then transferred to 37 °C for culture. EGFP fluorescence was monitored at 12, 24, 48, and 72 h postreactivation (hpr). (**F**) Detection of viral genomic DNA during reactivation. N2a cells were infected with rPRV-ΔUL54-EGFP and rPRV-ΔUL54-EGFP (MOI = 0.01). The viral DNA was extracted, and the *gB* gene was quantified at 0, 12, 24, 48, and 72 hpr. (**G**) Identification of expression levels of viral envelope proteins following rPRV-ΔUL54-EGFP reactivation. Bars represent the means ± SDs of three independent experiments. ns: not significant (*p*  ≥ 0.05). * *p* < 0.05, *** *p* < 0.001, and **** *p* < 0.0001.

**Table 1 viruses-15-00808-t001:** Primers used to construct the recombinant pseudorabies viruses.

Primers	Sequences (5′ to 3′)
US9-rpsl-F	AGAGCTGGTTTAGTGAACCGTCAGATCCGCTAGCGCTACCGGTCGCCACCGGCCTGGTGATGATGGCGGGATC
US9-rpsl-R	GGGCGCGGCGGATGGGGGCGGGCCCCCGCTCCCGTTCGCTCGCTCGCTCGTCAGAAGAACTCGTCAAGAAGGC
US9-EGFP-F	AGAGCTGGTTTAGTGAACCGTCAGATCCGCTAGCGCTACCGGTCGCCACCCGCTAGCGCTACCGGTCGCCAC
US9-EGFP-R	GGGCGCGGCGGATGGGGGCGGGCCCCCGCTCCCGTTCGCTCGCTCGCTCGATAACTTCGTATAGCATACATT
US9-JC-F	CGTATTAGTCATCGCTATTAC
US9-JC-R	AACAGAGACGCGGAGGAGAGG
UL54-rpsl-F	GGTTGCAGTAAAAGTACTTCCCGTGCATGTACACGGGGACGAGGGTGTAGGGCCTGGTGATGATGGCGGGATC
UL54-rpsl-R	AACAGCAGCGGCAGCGAGGCGTCCCGGTCGGGGAGCGAGGAGCGGCGCCCTCAGAAGAACTCGTCAAGAAGGC
UL54-del-F	GGTCTTGTGGGCGTGAGCCGCGCCCGGACGGGCGGC
UL54-del-R	TCCGGGCGCGGCTCACGCCCACAAGACCGGCTGCGA
UL54-JC-F	CTCGCGCACGCCAGAGAGGTAC
UL54-JC-R	CGCTCGCACCACGGTCATGGAG

**Table 2 viruses-15-00808-t002:** Primers for PCR, qPCR, and RT-qPCR.

Primers	Sequences (5′ to 3′)
EGFP-JC-F(PCR)	ATGGTGAGCAAGGGCGAGGAG
EGFP-JC-R(PCR)	TTACTTGTACAGCTCGTCCAT
gB-JC-F(PCR)	ATGGACATGTACCGGATCATGT
gB-JC-R(PCR)	AGAGCGTGACGCGCAACTTTCTGC
IE180-F(qPCR/RT-qPCR)	CTGGCAGAACTGGTTGAAGC
IE180-R(qPCR/RT-qPCR)	TCGTGCGCCTCATCTACAG
EP0-F(qPCR/RT-qPCR)	TGCGCCGATATGTCAAACAG
EP0-R(qPCR/RT-qPCR)	TCGTGGACAACATCGTCGAG
gB-F(qPCR/RT-qPCR)	TCCTCGACGATGCAGTTGAC
gB-R(qPCR/RT-qPCR)	ACCAACGACACCTACACCAAG
LAT-F(qPCR/RT-qPCR)	ACGTGACGTTTTTGCCGATG
LAT-R(qPCR/RT-qPCR)	GCGCGATATGCAGATGAGATC

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
