# Peer review of "Establishment of an In Vitro Model of Pseudorabies Virus Latency and Reactivation and Identification of Key Viral Latency-Associated Genes"

_viruses, 2023, doi:10.3390/v15030808_

Round 1

Reviewer 1 Report

In the current study titled “Establishment of an In Vitro Model of Pseudorabies Virus Latency and Reactivation and Identification of Key Viral Latency-Associated Genes”, Pan, al et. attempted to establish a powerful and streamlined model to simulate PRV latency and verify the vital role of temperature in PRV reactivation and disease. Moreover, the crucial role of the early gene UL54 in the latency and reactivation of PRV was addressed. Generally, this work is very interesting, the results are informative, the evidence is robust and the paper is well written. These findings are beneficial to the field.

Minor concerns include:

1.     A complete check for the writing including spelling is suggested

2.     To substantiate the evidence, at least a Western Blot analysis for the alteration of PRV viral protein in Figure 4 or 6 is suggested.

In the current study titled “Establishment of an In Vitro Model of Pseudorabies Virus Latency and Reactivation and Identification of Key Viral Latency-Associated Genes”, Pan, al et. attempted to establish a powerful and streamlined model to simulate PRV latency and verify the vital role of temperature in PRV reactivation and disease. Moreover, the crucial role of the early gene UL54 in the latency and reactivation of PRV was addressed. Generally, this work is very interesting, the results are informative, the evidence is robust and the paper is well written. These findings are beneficial to the field.

Minor concerns include:

1.     A complete check for the writing including spelling is suggested

2.     To substantiate the evidence, at least a Western Blot analysis for the alteration of PRV viral protein in Figure 4 or 6 is suggested.

Author Response

In the current study titled “Establishment of an In Vitro Model of Pseudorabies Virus Latency and Reactivation and Identification of Key Viral Latency-Associated Genes”, Pan, al et. attempted to establish a powerful and streamlined model to simulate PRV latency and verify the vital role of temperature in PRV reactivation and disease. Moreover, the crucial role of the early gene UL54 in the latency and reactivation of PRV was addressed. Generally, this work is very interesting, the results are informative, the evidence is robust and the paper is well written. These findings are beneficial to the field.

Author’s responses: We greatly appreciate Reviewer #1 for the positive comments and constructive suggestions.

Q1. A complete check for the writing including spelling is suggested

Author’s responses: We have revised the manuscript comprehensively, including typos, grammatical errors, lengthy sentences, etc.

Q2. To substantiate the evidence, at least a Western Blot analysis for the alteration of PRV viral protein in Figure 4 or 6 is suggested.

Author’s responses: We have included the Western blotting data in Figures 4 and 6 in the revised manuscript as suggested.

Reviewer 2 Report

The authpors report an in vitro model of PRV latency and reaction by high temprature. This is a useful model for the investigation of alphaherpesvirus latent infection. Overall, the quality of the study is good, and the data and experimental designs are convincing.

Minor comment:

Whether this in vitro  model of PRV latency and reaction by high temprature is occured in physiological conditions in vivo, such as fever?

Author Response

The authors report an in vitro model of PRV latency and reaction by high temperature. This is a useful model for the investigation of alphaherpesvirus latent infection. Overall, the quality of the study is good, and the data and experimental designs are convincing.

Author’s responses: We greatly appreciate the reviewer’s valuable comments and professional suggestions.

Q: Whether this in vitro model of PRV latency and reaction by high temperature is occured in physiological conditions in vivo, such as fever?

Author’s responses: It has been reported that PRV replicates rapidly in the mucosal or epithelial cells of pigs upon infection. Soon after, the intense inflammatory response induced by viral infection leads to a rapid rise in body temperature up to 42°C. In the presence of high temperature and host immune defense, residual virus enters the host peripheral nervous system to establish latency. A previous study showed that fewer or no infectious virus was produced in the BHK-21 cells infected with an attenuated PRV of a low MOI in 40°C, while virus activation occurred when the cells were transferred to 37°C. Recent studies have shown that HSV-1 is also able to maintain a latent state in human diploid fibroblasts or fetal lung cells cultured at 42°C. We speculate that this in vitro model of PRV latency and reaction by high temperature will be occurred in physiological conditions in vivo, but this needs to be confirmed in the following experiments, which were discussed in the revised manuscript.

References:

(1) Golais F, Sabó A, Rajcáni J. Latent Pseudorabies Virus Infection Established at Supraoptimal Temperature. Acta Virol. 1978, 22(6):464-469. PMID: 35944.

(2) Rafael D, Tomer E, Kobiler O. A Single Herpes Simplex Virus 1 Genome Reactivates from Individual Cells. Microbiol. Spectr. 2022, 10(4):e0114422. PMID: 35862979.
